# Effects of Two Feeding Patterns on Growth Performance, Rumen Fermentation Parameters, and Bacterial Community Composition in Yak Calves

**DOI:** 10.3390/microorganisms11030576

**Published:** 2023-02-24

**Authors:** Qin Li, Yan Tu, Tao Ma, Kai Cui, Jianxin Zhang, Qiyu Diao, Yanliang Bi

**Affiliations:** 1Institute of Feed Research, Chinese Academy of Agricultural Sciences/Sino-US Joint Lab on Nutrition and Metabolism of Ruminant/Key Laboratory of Feed Biotechnology of the Ministry of Agriculture and Rural Affairs, Beijing 100081, China; 2College of Animal Science, Shanxi Agricultural University, Jinzhong 030801, China

**Keywords:** yak, yak calf, weaning, rumen, milk replacer

## Abstract

The health of young ruminants is highly dependent on early rumen microbial colonization. In this study, the effects of milk replacer on growth performance, rumen fermentation, and the rumen microflora in yak calves were evaluated. Sixty yak calves (body weight = 22.5 ± 0.95 kg, age = 30 ± 1 d) were assigned to the CON group (breastfed) or TRT group (milk replacer fed) and evaluated over 120 d. At 120 d, ruminal fluid samples were collected from 14 calves and then conducted for rumen fermentation and microbiota analyses. There was no difference in growth performance; however, calf survival was higher in the TRT group than in the CON group. The concentration of total volatile fatty acids and the molar proportion of butyric acid and lactic acid were increased with milk replacer feed in the TRT group (*p* < 0.05), but iso-valeric acid concentration was highest in the CON group (*p* < 0.05). *Firmicutes* and *Bacteroidetes* were the most dominant phyla in the CON and TRT groups, respectively. In the TRT group, *Bacteroidetes*, *Prevotellaceae*, *Bacteroidia*, *Bacteroidetes*, and *Prevotella_1* were the dominant flora in the rumen of calves. The relative abundances of various taxa were correlated with rumen fermentation parameters; the relative abundance of *Quinella* and iso-butyrate levels were positively correlated (*r* = 0.57). The relative abundances of the Christensenellaceae_R-7_group and A/P were positively correlated (*r* = 0.57). In summary, milk replacer is conducive to the development of the rumen microflora, the establishment of rumen fermentation function, and the implementation of early weaning in yaks.

## 1. Introduction

The yak (Bos grunniens) is an important species on the Tibetan Plateau (3000 m above sea level) [1]. This ruminant is a “living treasure” for the Tibetan people because it can provide milk, meat, hair, hides, and feces (a valuable fuel) [2,3]. The nutritional status of the mother-calf is the most important determinant of female yak reproduction, and farms tend to shorten the duration of the postpartum off-cycle by weaning early and restricting the nutrition provided by cows to calves, which results in slower growth and higher winter mortality in newborn calves [4]. The reproductive rate of adult female yaks undergoing conventional breastfeeding is only 48.61% in this environment [5]. In addition, the survival rate of newborn calves is meager due to extreme cold weather and malnutrition [6,7]. As a result, yak breeding is highly challenged.

Milk replacer (MR) has shown beneficial effects in animal experiments [8] and has a positive effect on animal growth and gastrointestinal development [9,10]. MR can not only effectively replace breastfeeding and help the mother and offspring recover their physical condition as soon as possible but can also help the ruminal microorganisms of the calf adapt earlier during the transition from liquid feed to solid feed [11] and promote the colonization of dominant bacteria in the rumen. In ruminants, rumen fermentation must be considered in the feeding management period from birth to weaning [12]. Early feeding programs and nutrition significantly impact the diversity and evolution of the rumen microbial community [13,14]. The early colonization of rumen microorganisms is significant for rumen fermentation and later growth and development [15]. The effect can last for a long time and therefore affect the performance and health of adult ruminants for life. Concerning underlying mechanisms, we focused on the effect of MR consumption in yak calves on rumen fermentation and the rumen microflora in this study.

To determine whether MR can achieve the same effect as breastfeeding in yak calves, growth performance, rumen fermentation, and microbiota were evaluated. We hypothesized that MR could promote growth, rumen fermentation, and microbial colonization in yak calves. The results of this study provide a theoretical basis for the use of MR for early weaning in yak calves.

## 2. Materials and Methods

The experiment protocol was approved by the ethics committee of the Chinese Academy of Agricultural Sciences Animal, and it was performed in accordance with the animal welfare practices and procedures in the Guidelines for Experimental Animals of the Ministry of Science and Technology.

### 2.1. Animals and Experimental Design

A total of 60 yak calves (body weight = 22.5 ± 0.95 kg, age = 30 ± 1 d) were randomly divided into control (CON) and experimental (TRT) groups, with 30 calves in each group. In the CON group, yak calves were breastfed and lived with their mothers throughout the experiment. In the TRM group, yak calves were separated from female yaks and fed on MR. The experiment was conducted for a total of 120 days in Nagqu, Tibet, from September to December 2019.

MR for yak calves (Patent Number: 02128844.5) was obtained from the Beijing Precision Animal Nutrition Research Center, China. The nutrient composition of the MR is shown in Table 1. Boiling water was cooled to 50 °C and MR was added at a MR:water ratio of 1:7 until the emulsion cooled naturally to 39 °C. MR was bottle-fed to calves in the TRT group twice a day (08:00 and 18:00) at 1.5% (DM) of the body weight. All calves were allowed to drink water ad libitum.

### 2.2. Growth Performance and Survival Rate

At days 0 and 120, yak calves were weighed, and the average daily gain was calculated. During the experiment, the mortality rate of calves was recorded, and the survival rate in each group was calculated. 

### 2.3. Milk Replacer Nutrient Content

Dry matter (DM) content was determined by drying in an oven at 105 °C for 6 h. Crude protein (CP) was determined by the Kjeldahl method (FOSS-8400). Ether extract (EE) was obtained by Soxhlet extraction. The ash content was determined by a muffle furnace burning for 6 h. Ga was determined by atomic spectrophotometry (M9W-700; Perkin-Elmer, Waltham, MA, USA). The P content was evaluated by molybdenum vanadate colorimetric determination.

### 2.4. Ruminal Chyme Collection 

On d120, seven calves in each group were used for the collection of ruminal chyme samples through the oral cavity. The first extracted chyme was discarded, and then ruminal chyme samples were aspirated from different parts and immediately frozen in a −80 °C liquid nitrogen tank after mixing completely. Subsequently, microbiome and rumen fermentation characteristics were analyzed.

### 2.5. Ruminal Analysis of Fermentation Parameters

After samples were thawed, they were centrifuged at 20,000× *g* and 4 °C for 15 min. Three 1 mL rumen fluid samples were extracted, 0.25 mL of metaphosphoric acid was added, and volatile fatty acid (VFA) profiles were determined using a gas chromatography instrument (GC-6800; Beijing Beifen Tianpu Instrument Technology Co., Ltd., Beijing, China). 

### 2.6. DNA Extraction, PCR Amplification, and 16S rRNA Sequencing

Microbial DNA was extracted from rumen fluid samples using the OMEGA E.Z.N.A.^®^ Digesta DNA Kit (OMEGA Bio-Tek, Norcross, GA, USA) according to the protocol described by the manufacturer. DNA quality and quantity were determined using a ND 1000 spectrophotometer (NanoDrop Technologies Inc., Wilmington, DE, USA). Using primers 338F (5′-ACTCCTACGGGAGGCAGCAG-3′) and 806R (5′-GGACTACHVGGGTWTCTAAT-3′), the V3-V4 region of the bacterial 16S ribosomal RNA gene was amplified by PCR. The conditions for the PCR were as follows: 95 °C for 3 min, followed by 95 °C for 30 s, 55 °C for 30 s, 72 °C for 45 s, and finally 72 °C for 10 min. The barcode sequence was eight bases long and unique to each sample. The PCR was performed in three 20 μL mixtures consisting of 4 μL of 5× FastPfu Buffer, 2 μL of 2.5 mM dNTPs, 0.8 μL of each primer (5 μm), 0.4 μL of FastPfu Polymerase, and 10 ng of template DNA. Amplicons were extracted from 2% agarose gels and purified using the AxyPrep DNA Gel Extraction Kit (Axygen Biosciences, Union City, CA, USA) according to the manufacturer’s instructions. The QuantiFluor™-ST system (Promega, Madison, WI, USA) was used for quantification. Purified amplicons were aggregated in equimolar ratios on an Illumina MiSeq PE300 platform for 2 × 300 bp paired-end sequencing according to standard protocols.

### 2.7. Sequencing Data Processing

After demultiplexing the original FASTQ files, the sequences were filtered using Trimmomatic according to the following criteria: (i) within a 50 bp sliding window, reads longer than 300 bp were truncated at the site with a quality score of <20, and those shorter than 50 bp were discarded; (ii) reads containing any mismatches in the barcode region, two or more nucleotide mismatches in the primer sequence, or ambiguous characters were discarded; and (iii) only overlaps and GT; Bp and 10 and lt; 10% of mismatched sequences were assembled; unassembled reads were discarded. The assembled sequences were then trimmed with primers and barcodes. Chimeric sequences were identified and deleted using USearch. After quality control, assembled sequences were assigned to operational taxonomic units (OTUs) using UPARSE with a 97% identity threshold. RDP classifier version 2.2 and the Silva database were used for annotation at the domain, phylum, class, order, family, genus, and species levels. QIIME 2 was used to calculate the alpha diversity indexes, including the ACE, Chao1, Shannon, and Simpson indexes. The alpha indexes were compared between groups using Kruskal–Wallis tests in R version 4.0.2. The differences in bacterial communities between groups were evaluated by a principal coordinate analysis (PCoA) based on the Bray–Curtis dissimilarity matrix using QIIME 2. Linear discriminant analysis effect size was used to identify significant bacteria in the two groups. 

### 2.8. Statistical Analyses

All data were tested for normality using the Sharpiro-Wilk test in R-Studio (3.6.1) and were normally distributed (*p* > 0.05). Differences in growth performance and rumen fermentation parameters between the CON and TRT groups were compared using the unpaired *t*-test. 

Differences in alpha diversity and relative abundance at the phylum, family, and genus levels were evaluated [16]. The Kruskal–Wallis test (in R version 4.0.3) was used to compare the microbiota function between groups. The “ape”, “ggplot2”, “limma”, “GGplot2”, and “GGTree” packages in R were used to visualize results (i.e., to obtain PCoA plots, Venn diagrams, bar plots, and LEfSe plots) [16,17,18].

The “corrplot” package in R was used to analyze the correlations between the top 20 genera of all samples and rumen fermentation parameters (based on Spearman’s correlation coefficients). The network containing the top 20 genera was visualized using the “igraph” package in R version 4.0.3. The transformed data for the abundances of communities at the phylum and genus levels were analyzed by one-way ANOVA using SAS. The MIXED procedure model included the fixed effects of treatment and interactions between treatments, as well as the random effect of the individual nested within treatment. Treatment differences with *p* < 0.05 were considered statistically significant, and 0.05 ≤ *p* < 0.10 indicated marginal significance.

## 3. Results

### 3.1. Growth Performance and Survival of Yak Calves under Two Feeding Modes

As summarized in Table 2, there was no significant difference in growth performance between the two groups, indicating that MR satisfied the basic growth needs of calves. However, the survival rate of calves in the TRT group (73.33%) was significantly higher than that in the CON group (46.47%) (*p* < 0.05). This suggested that MR feeding helped calves adapt to the highland environment.

### 3.2. Rumen Fermentation Parameters for Yak Calves under Two Feeding Modes

As shown in Table 3, rumen fermentation parameters, particularly TVFA, butyric acid, iso-valeric acid, and lactic acid levels, differed significantly between the two feeding patterns (*p* < 0.05). The TRT group exhibited higher TVFA and lactic acid concentrations and higher molar ratios of butyric acid and lactic acid than those of the CON group (*p* < 0.05). The CON group had a higher molar ratio of iso-valeric acid than the TRT group (*p* < 0.05). The remaining fermentation parameters were similar between the two groups; there was no statistical significance.

### 3.3. Composition of Rumen Bacteria

#### 3.3.1. Overview of Sequencing Data for Rumen Microorganisms

The flora in rumen fluid samples was subjected to paired-end sequencing, and raw data were filtered to obtain 2,871,736 high-quality reads, with an average of 191,450 reads per sample. Based on the 97% sequence similarity threshold, 4182 OTUs were obtained from 15 groups of rumen fluid samples, of which 3679 OTUs were detected in the CON group and 3270 OTUs were detected in the TRT group. In total, 2785 (66.59% of the total OTUs) were common to both groups (Figure 1A). The sequencing coverage in both groups was 98% (Table 4), indicating that the depth of the sequencing reflects the true microbial profile in the rumen fluid of calves. The observed_species and PD_whole_tree in the CON group were significantly higher than those in the TRT group (*p* < 0.05), indicating a higher species abundance in the CON group. There were no significant differences in other diversity indices, indicating that the difference in microbial diversity between the two groups was small. The results of a PCoA (Figure 1B) showed that there was only partial overlap in OTU frequencies between the two groups, with significant differences (*p* < 0.003), as well as significant differences among individuals within the CON group.

#### 3.3.2. Relative Abundance of Bacterial Populations

To further evaluate the effect of MR on the rumen flora of calves, we compared the relative abundance of each taxon at the phylum and genus levels (Figure 2). *Firmicutes* (46.85%), *Bacteroidetes* (46.83%), and *Actinobacteria* (1.59%) were the dominant flora in the rumen of calves in both feeding modes. However, the most abundant bacterial phylum was *Bacteroidetes* (58.33%; *p* = 0.001) in the TRT group and *Firmicutes* in the CON group (55.47%; *p* = 0.003) (Figure 2A and Appendix A).

At the genus level, the results for both groups were similar to those for the phylum level, which suggests that the feeding pattern was one of the key drivers of the early colonization of rumen microbes. Notably, the top six dominant genera identified were common to both groups. However, their relative abundances differed between groups. In the TRT group, *Prevotella_1* (39.89%), *Christensenellaceae_R-7_group* (6.00%), *Christensenellaceae_R-7_group* (6.00%), the *Rikenellaceae_RC9_gut_group* (4.17%), *Succiniclasticum* (2.73%), and *Ruminococcaceae_NK4A214_group* (2.54%) were most abundant. The relative fractions of these bacteria in the CON group were 15.04%, 18.98%, 15.78%, 6.39%, 5.23%, 3.72%, and 2.62%, respectively. The relative abundances of the genera *Prevotella_1* (*p* = 0.001) and *Lachnospiraceae_XPB1014_group* (*p* = 0.005) were significantly higher in the TRT group than in the CON group. The relative abundances of *Prevotellaceae_UCG-003* (*p* = 0.026) and *Eubacterium_coprostanoligenes_group* (0.016%) were significantly lower in the TRT group than in the CON group (Figure 2B and Appendix A).

#### 3.3.3. LEfSe Analysis

To better identify specific bacteria in both groups, we used a LEfSe analysis and LDA scores (Figure 3). Taxa with LDA scores of >4 were identified as biomarkers. In the TRT group, the *Bacteroidetes* phylum, *Prevotellaceae* family, *Bacteroidia* class, *Bacteroidetes* order, and *Prevotella_1* genus were highly discriminative (LDA score >4.8). *Firmicutes* (LDA score > 4.8) was highly discriminative in the CON group.

### 3.4. The Correlation of Ruminal Fermentation Parameters with Bacterial Communities

Our results revealed that the rumen microbial composition differed significantly between the two groups. Such differences were often due to differences in the internal environment caused by dietary differences. Accordingly, we analyzed the correlations between the relative abundance of the top 20 bacterial genera and fermentation parameters (Figure 4).

The relative abundance of *Quinella* and iso-butyrate levels were positively correlated (*r* = 0.57). The relative abundances of the Christensenellaceae_R-7_group and A/P were positively correlated (*r* = 0.57). *Butyrivibrio*_2 relative abundance and iso-butyrate levels were negatively correlated (*r* = 0.57). The relative abundance of Veillonellaceae_UCG-001 was negatively correlated with TVFA (*r* = −0.55). The relative abundance of *Lachnospiraceae_XPB1014_group* was positively correlated with butyrate (*r* = 0.58). The relative abundance of *Ruminococcaceae_UCG-014* was positively correlated with lactate levels (*r* = 0.59). The relative abundance of *Saccharofermentans* was positively correlated with A/P (*r* = 0.58). The relative abundance of *Ruminococcus_1* was negatively correlated with propionate levels (*r* = −0.55) and positively correlated with A/P (*r* = 0.60) (Figure 4A). The relative abundance of *Lachnospiraceae_NK3A20_group* and iso-butyrate levels had a highly significant positive correlation (*r* = 0.76) (Figure 4B).

## 4. Discussion

The altitude at which this experiment was carried out (4436 m) was higher than that of a previous study (approximately 3000 m) [19]. However, weight gain data for 30-day-old yak calves have not been reported in existing literature. Yak calves could be weaned and fed MR as early as 30 days after birth; early weaning improved yak calves’ survival rate. 

We found that MR can completely replace yak milk to feed yak calves; this study provides the first evidence supporting the use of MR in yak calves. Ruminal fermentation parameters are potential markers of rumen microorganisms and are representative of how rumen microorganisms respond to diet [20,21,22]. Ruminal VFA can provide up to 70% of the total energy requirements for ruminants [21]. The total volatile acid concentration was significantly higher in the TRT group than in the CON group, which may be related to a difference in the total intake of dry matter [23]. Furthermore, calves in the TRT group exhibited steady feed intake on a daily basis throughout the experiment. Yak lactation decreases once the calf reaches 2–6 months of age [24]. Therefore, we thought that feed intake was affected by this decrease in the CON group [25]. Butyrate is one of the three most highly produced VFAs in the rumen and has beneficial effects on calf gastrointestinal development, pancreatic secretion, and nutrient digestion [26,27,28]. The incidence of diarrhea is reduced by butyrate [29]. Lactose is a candidate for butyrate enhancement in the rumen, as it increases the concentration of butyrate in cows [30] and promotes the proliferation of probiotics such as *Lactobacillus* and *Bifidobacterium* [31]. On the other hand, lactose provided more fermentation substrate for microorganisms in the TRT group. Iso-valerate is a branched-chain fatty acid produced by the rumen fermentation of branched-chain amino acids [32]. It could increase the number of fibrinolytic bacteria with little effect on calves during lactation [33]. A total of 40%–68% of the proteins synthesized by microorganisms are derived from NH_3_-N [34]. In this study, the differences in NH_3_-N concentrations between the two groups were not significant, but the results in two groups were in the concentration range with the highest efficiency (5–25 mg/dL) [35]. TRT provided more nitrogen, promoted the synthesis of MCP, and resulted in more metabolizable proteins in the small intestine for use by calves. Rumen fermentation results were consistent with those of a previous study involving yak calves (TVFA = 68 mmol/L) [36]; however, in 6-month-old calves eating roughage, the NH_3_-N concentrations were reduced (NH_3_-N = 5.46 mg/dL) [36]. Considering these results, we believe that MR acts as a stable and rich nutrient source for yaks, providing a favorable environment for the early colonization of rumen bacteria.

At 3 weeks of age, concentrate feed intake allows calves to transition from functional non-ruminants to true ruminants, a process that relies on the establishment and activity of the rumen microbiota. Until weaning, the rumen is fully functional [37]. However, the microbial taxa commonly found in the mature rumen are established in the rumen of calves [38,39]. MR ensured that the calves had more concentrated diets in the TRT group, which led to a decrease in the rumen microbiota [40], and this was likely related to the high content of dominant bacteria. *Firmicutes* and *Bacteroides* are the main phyla in the rumen, intestines, and feces, and *Firmicutes* is more dominant in the yak rumen [41,42]. This dominance is positively correlated with the fiber degradation capacity, which is conducive to the digestion of poor-quality pasture [43]. We discovered an increase in microbial diversity of domestic yaks, which enriches *Firmicutes* in wild yaks [44]; *Firmicutes* contributes to effective energy harvesting [45]. Calves of the TRT group likely had advantages in adapting to the harsh environment of the plateau. *Verrucomicrobia* is the dominant bacteria in the rumen of grazing yaks in the CON group, and the high abundance of this taxon may be due to grazing [46].

The relative abundance of *Prevotella* is high in yak rumen [47], and another study indicated that *Prevotella_1* is most abundant in yak stomachs and duodenum [46]. The genus *Prevotella* is known for its high VFA production using rumen proteins, which may also explain why *Prevotella_1* was the most dominant bacteria in both groups [47]. However, calves in the TRT group had higher TVFA contents than those in the CON group. *g__Quinella* is an H_2_-incorporating bacterium associated with low methane production [48]. The *g__Christensenellaceae_R-7_group* belongs to the phylum *Firmicutes* and is associated with the breakdown of proteins in feed and can promote the absorption of nutrients. The *g__Rikenellaceae_RC9_gut_group* is abundant in yaks and may play a role in the digestion of plant-derived polysaccharides [49]. These genera are conducive to rumen digestion and absorption and did not differ significantly between the two groups. In summary, the microbial structure of the TRT group was conducive to the absorption and degradation of nutrients, emphasizing the beneficial effects of MR.

We analyzed the correlations between the top 20 bacteria and rumen fermentation parameters, revealing that nine bacterial taxa had obvious relationships with rumen fermentation parameters. These bacteria were mainly related to A/P and isobutyric acid concentrations. This suggests that iso-butyric acid and A/P play important roles in the establishment of rumen function before weaning. Iso-butyric acid is a degradation product of proteins in the rumen, and a low concentration of it affects the growth and reproduction of microorganisms [50]. It was reported that the concentration of acetate in the rumen increased with isobutyric acid added to the diet in beef cattle, the concentration of propionic acid remained unchanged, and A/P increased [51]. It was found that the *Lachnospiraceae_NK3A20_group* was positively correlated with the butyric acid concentration in dairy cows with early lactation [52]. A/P is related to energy utilization and can affect the microbiota structure in the rumen, which is crucial for early breeding [53]. Guo et al. And it was found that *Dioscorea* feeding increases the A/P ratio and increases the abundance of *g__Ruminococcus_1* [54]. Additionally, *g__Saccharofermentans* is a probiotic and is inversely correlated with propionate concentration [55]. Therefore, we believed that MR optimized the structure of the rumen flora and promoted rumen fermentation by regulating rumen fermentation products.

We only discuss rumen microbes but not related studies of gut microbes and metabolomics in the present study, and also due to limited experimental conditions, yak calves were not observed for long enough, especially in the response of yak calves to solid diet after weaning, which would be worth more investigation in the future.

## 5. Conclusions

Substituting MR completely replaced yak milk to feed yak calves, optimized the rumen microbiota structure, and ameliorated rumen fermentation by influencing fermentation products, thereby improving the survival rate of yak calves. These results support the positive impact of MR on early weaning and breeding in yaks. It provides an opportunity for the improvement of the yak calf survival rate and the expansion of farmers’ economic profits. 

## Figures and Tables

**Figure 1 microorganisms-11-00576-f001:**
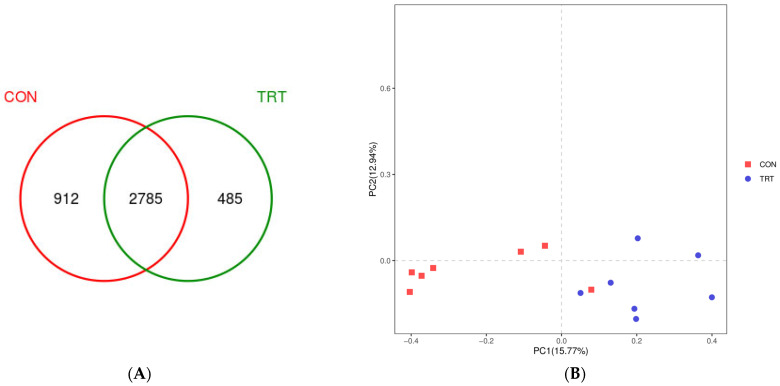
Response of the rumen microbiota to two feeding patterns. (**A**) Microbial composition of yak calves at the OTU level (Venn diagram). (**B**) PCoA of the rumen microbiota of yak calves after two feeding patterns. CON: breastfeeding; TRT:milk replacer feeding.

**Figure 2 microorganisms-11-00576-f002:**
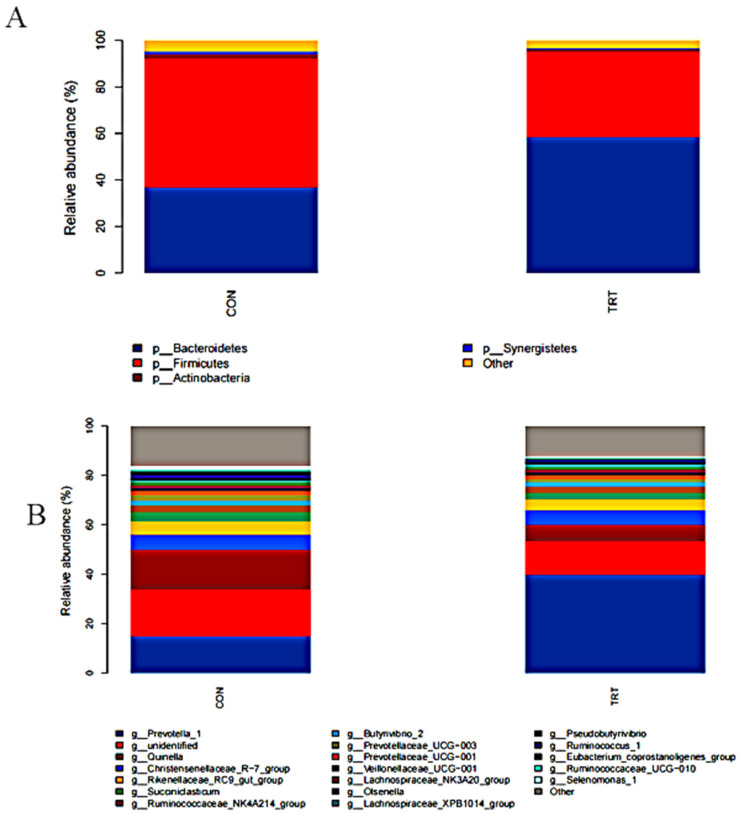
Compositions of the yak calf rumen microbiota under two feeding patterns. (**A**) Phylum level; (**B**) Genus level. CON: breastfeeding; TRT:milk replacer feeding.

**Figure 3 microorganisms-11-00576-f003:**
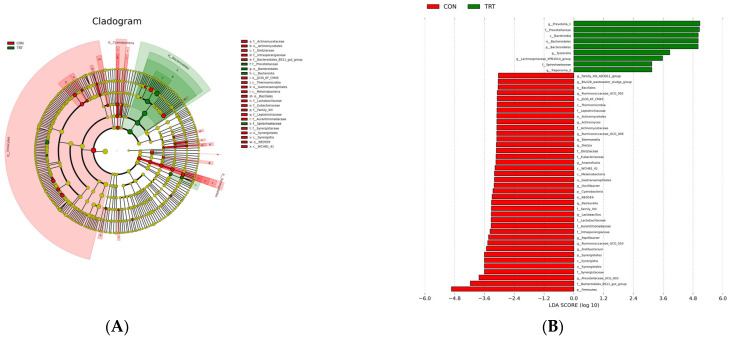
LEfSe analysis of the rumen microbiota from yak calves with two feeding patterns. (**A**) Histogram of linear discriminant analysis scores based on classification information. (**B**) Linear discriminant analysis effect size cladogram based on classification information. CON: breastfeeding; TRT:milk replacer feeding.

**Figure 4 microorganisms-11-00576-f004:**
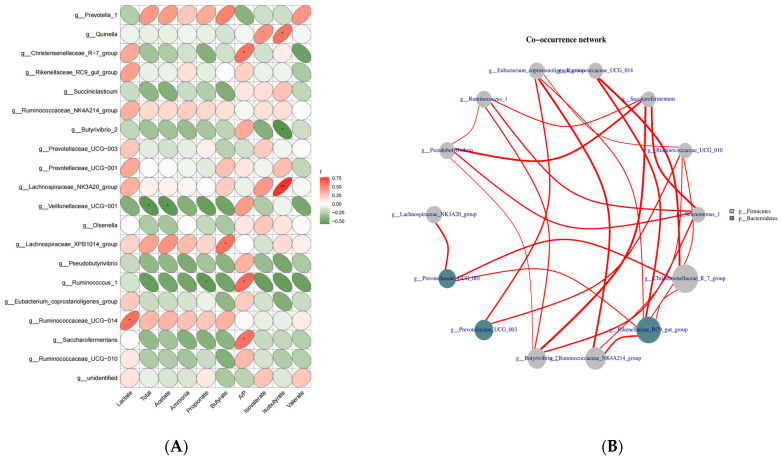
Correlation analysis of rumen genera from yak calves under two feeding patterns. (**A**) Correlations between the relative abundances of the top 20 genera with fermentation parameters. (**B**) Interactions among genera with the top 20 relative abundances. Red represents a positive correlation, and blue represents a negative correlation. The number indicates the correlation coefficient. * *p* > 0.05, ** *p* > 0.01. The size and color of the circle represent the relative abundance of the respective genus and phylum. CON: breastfeeding; TRT:milk replacer feeding.

**Table 1 microorganisms-11-00576-t001:** Nutrient levels in the milk replacer (descriptive summary of feed composition; analyzed values indicate % of DM product unless specified otherwise).

Nutrient ^1^	Milk Replacer ^2^ Composition (%)
DM	94.0
CP	24.0
EE	16.0
Ash	10.0
Ga	2.0
P	0.9
NaCl	1.0
Lysine	2.2
Vitamin E, IU	70.0

^1^ DM, dry matter; CP, crude protein; EE, ether extract; Ash, crude ash. ^2^ The main ingredients of milk replacer: whole milk powder, whey powder, protein concentrate, vitamin A, vitamin D3, vitamin E niacin, pantothenic acid, lysine, methionine, threonine, ferrous glycinate, zinc glycinate, etc.

**Table 2 microorganisms-11-00576-t002:** Effects of two feeding patterns on the growth performance and survival rate of yak calves.

Items	Groups ^1^		SEM	*p*-Value
	CON	TRT		
n1	30	30		
n2	14	22		
Survival rate,%	46.47	73.33		
IBM, kg	22.71	22.34	0.89	0.819
FBM, kg	43.73	43.84	1.21	0.924
ADG, g/d	165.23	159.94	6.77	0.529

^1^ CON group: breastfeeding; TRT group: milk replacer feeding. n1: sample number at the beginning of the experiment in each group; n2: sample number at the end of the experiment in each group; survival rate (%) = n2/n1; ADG, average daily gain (g/d) = FBM, final body weight-IBM, initial body weight/Number of days.

**Table 3 microorganisms-11-00576-t003:** Effects of two feeding patterns on rumen fermentation parameters in yak calves.

Items	Groups ^1^		SEM	*p*-Value
	CON	TRT		
n	7	7		
TVFA, mmol/L	50.19	79.18	6.03	0.017
Molar proportion				
Acetate, %	65.87	62.96	1.15	0.252
Propionate, %	20.28	22.19	1.21	0.482
Butyrate, %	8.52	10.93	0.50	0.016
Iso-butyrate, %	1.96	1.29	0.19	0.095
Valerate, %	1.04	1.32	0.13	0.342
Iso-valerate, %	2.32	1.31	0.20	0.013
A/P	3.5	2.92	0.27	0.220
NH3-N concentration, mg/dL	10.10	18.10	2.61	0.082
Lactate, mmol/dL	3033.20	3944.48	182.80	0.021

^1^ CON group:breastfeeding; TRT group:milk replacer feeding. TVFA: total volatile fatty acids; A/P: acetic acid/propionic acid.

**Table 4 microorganisms-11-00576-t004:** Effects of different feeding patterns on rumen microbial alpha diversity in yak calves.

Items	Group 1		SEM	*p*-Value
	CON	TRT		
Chao1	2335.36	2059.59	272.70	0.055
Goods_coverage,%	0.98	0.98	0.00	0.847
Observed_species	1845.26	1534.2	275.24	0.029
PD_whole_tree	155.41	134.27	18.75	0.029
Shannon	7.96	7.49	1.18	0.48
Simpson	0.95	0.97	0.06	0.51

^1^ CON group: breastfeeding; TRT group:milk replacer feeding.

## Data Availability

The data that support the findings of this study are available on request from the corresponding author. The data are not publicly available due to privacy or ethical restrictions.

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
