# Peer review of "Effects of Two Feeding Patterns on Growth Performance, Rumen Fermentation Parameters, and Bacterial Community Composition in Yak Calves"

_microorganisms, 2023, doi:10.3390/microorganisms11030576_

Round 1

Reviewer 1 Report

The language style is poor, incomprehensible and confusing.

Line 12 : changed to milk replacer

Line 15 : Changed “compared by gas chromatography and high-throughput sequencing” to rumen fermentation and microbiome analyses. When did you take rumen liquor?

Line 17-18: Please reorganize the sentence : Total volatile fatty acids, butyric acid, and lactic acid levels were higher in the TRT group than in the CON group, and iso-valeric acid levels were highest in 18 the CON group (all P < 0.05).

Line 19: Delete “Microbial compositions differed between groups. For example,

Line 20-22: Please reorganize the sentence, y. In the TRT group, Bacteroidetes, Prevotellaceae, Bacteroidia, Bacteroidetes, and Prevotella_1 were discriminative. Firmicutes was discriminative in the CON group

Line 23 : changed to milk replacer

Line 29 : Changed “This ruminant is vital to the Tibetan people” to due to

Line 30-35: Please reorganize the sentence “Female yaks gen erally calf in May. Under conventional breastfeeding [4], the reproductive rate of adult female yaks is only 48.61% [5]. Yak germplasm resources have been degraded, and seasonal malnutrition is one of the main causes [6]. This seasonal variation not only affects growth but also seriously affects the survival of newborn yaks [7]. Accordingly, yak breeding is highly challenging

Line 27-63 : The language style is incomprehensible. You should improve your english and writing style

Line 71 : are the animals in the same number (between groups) ? What does matched for weight mean?

Line 82: Could you mention the compositions?

Line 167 : The figures are not clear

Line 198 : The figures are not clear

Line 214-215: Could you make the figure of genus and phylum level in groups, instead of individuals? The figures are not clear

Line 238 : The figures are not clear

Line 249 : The figures are not clear

Author Response

Reviewer: 1

Comments to the Author

MAJOR CORRECTIONS:

Point 1: The language style is poor, incomprehensible and confusing.

Response 1: This manuscript has been touched up by a professional language company and the results were inserted in the attachment. We marked in red in the revised manuscript.

Point 2: Line 12 : changed to milk replacer

Response 2:  "Milk Replacer" was replaced by "milk replacer" in line 12 and marked in red as suggested in the revised manuscript.

Point 3: Line 15 : Changed “compared by gas chromatography and high-throughput sequencing" to rumen fermentation and microbiome analyses. When did you take rumen liquor?

Response 3: The change was marked in red in the revised manuscript (Lines 15-16). We sampled rumen liquor d120 of the experiment.

Point 4: Line 17-18: Please reorganize the sentence : Total volatile fatty acids, butyric acid, and lactic acid levels were higher in the TRT group than in the CON group, and iso-valeric acid levels were highest in 18 the CON group (all P < 0.05).

Response 4: In the revised manuscript, we modified the expression of the results on total volatile fatty acidsto make the presentation more concise and clearer (lines 17-19).

Point 5: Line 19: Delete “Microbial compositions differed between groups. For example,

Response 5: It was deleted as suggested in the revised manuscript (line 19).

Point 6: Line 20-22: Please reorganize the sentence, y. In the TRT group, Bacteroidetes, Prevotellaceae, Bacteroidia, Bacteroidetes, and Prevotella_1 were discriminative. Firmicutes was discriminative in the CON group

Response 6: In the revised manuscript, we found “discriminative" was not very accurate, we have changed it to “ were the dominant flora in the rumen of calves " (Line 22).

Point 7: Line 23 : changed to milk replacer

Response 7: The change was marked in red in the revised manuscript (Line 23).

Point 8: Line 29 : Changed “This ruminant is vital to the Tibetan people” to due to

Response 8: The change was marked in red in the revised manuscript (Line 30).

Point 9: Line 30-35: Please reorganize the sentence “Female yaks generally calf in May. Under conventional breastfeeding [4], the reproductive rate of adult female yaks is only 48.61% [5]. Yak germplasm resources have been degraded, and seasonal malnutrition is one of the main causes [6]. This seasonal variation not only affects growth but also seriously affects the survival of newborn yaks [7]. Accordingly, yak breeding is highly challenging

Response 9: In the revised manuscript, we refined the introduction section order to make it more logical. The change was marked in red (Line 29-36).

Point 10: Line 27-63 : The language style is incomprehensible. You should improve your english and writing style

Response 10: In the revised manuscript, we cut out some unnecessary content for clarity and added new literature to support, we made changes to the introduction based on your suggestions (Line 29-57).

Point 11: Line 71 : are the animals in the same number (between groups) ? What does matched for weight mean?

Response 11: Yes, the number of two groups were the same. In order to avoid the influence of body weight on the test results, the weight of the test animals was controlled at 22.5 ± 0.95 kg, and then they were randomly grouped to ensure that there was no difference between the two groups. The original manuscript has deleted the incorrect use of "matched". 

Point 12: Line 82: Could you mention the compositions?

Response 12: The main ingredients of this milk replacer involve soy and other plant proteins, whey powder, water-soluble fats, minerals and vitamins as raw materials. The nutrient content in the manuscript were all measured values.

Point 13: Line 167 : The figures are not clear

Line 198 : The figures are not clear

Line 238 : The figures are not clear

Line 249 : The figures are not clear

Response 13: In the revised manuscript, the original figures were used to ensure the clarity, especifically revised to lines 196, 235 and 245. We also changed Figure 1 to Table 2 in line 165.

Point 14: Line 214-215: Could you make the figure of genus and phylum level in groups, instead of individuals? The figures are not clear

Response 14: We modified the results for the phylum and genus levels in the figure to compare between two groups, also changed the figures format to make they more clearer in line 209 of the revised manuscript. 

Reviewer 2 Report

To conduct Yak experiments are difficult, especially in the Nagqu region of Tibet, China, where the altitude is very high and data are not easy to obtain;

The article is well organized and also provides a means for reducing mortality in yak calves.

Suggestion:

L24 The abstract says that it promotes rumen development, how did you get this conclusion without slaughtering calves?

A 22kg calf is no longer newborn calves, how is it raised and managed in after birth of the two groups ?

Table 1 Does MR have a vitamin and trace elements content ?

L96 If rumen fluid is taken only once, how to ensure uniformity?

L143 How can there be multiple comparisons between 2 groups ? How was the mortality or survival anzlyzed ?

TABLE 2  why did rumen fermentation characteristics not include MCP and PH?

3.4 relationship or correlation ?

Author Response

Reviewer: 2

Comments to the Author

To conduct Yak experiments are difficult, especially in the Nagqu region of Tibet, China, where the altitude is very high and data are not easy to obtain;

The article is well organized and also provides a means for reducing mortality in yak calves.

Point 1: L24 The abstract says that it promotes rumen development, how did you get this conclusion without slaughtering calves?

Response 1: “Rumen development” may not be accurately stated in this manuscript, and we changed “rumen development” to “rumen fermentation”.(lines 25, 54, 125 and 333). Rumen development may not necessarily require slaughter, and rumen morphology, rumen fermentation parameters, and rumen microorganisms can all reflect the effects on rumen development. In most of the experiment involving cattle, because the cost of slaughter is very high, they rated rumen development by rumen fermentation parameters and rumen microorganisms. Here are some published articles.

[1].Parsons, S.D., et al., Effect of a milk byproduct–based calf starter feed on dairy calf nutrient consumption, rumen development, and performance when fed different milk levels. Journal of Dairy Science, 2022. 105(1): p. 281-300. https://doi.org/10.3168/jds.2021-21018

[2].Zhao, C., et al., Yak rumen microbiome elevates fiber degradation ability and alters rumen fermentation pattern to increase feed efficiency. Animal Nutrition, 2022. 11: p. 201-214. https://doi.org/10.1016/j.aninu.2022.07.014

[3].GUO, Y., et al., Effect of Dioscorea Opposite Waste on Growth Performance, Blood Parameters, Rumen Fermentation and Rumen Bacterial Community in Weaned Lambs. Journal of Integrative Agriculture, 2022. https://doi.org/10.1016/j.jia.2022.10.002

Point 2:A 22kg calf is no longer newborn calves, how is it raised and managed in after birth of the two groups ?

Response 2: Yak calves were born in August, and the newborn calves were not able to withstand the coming cold snap, so we chose 30-day-old calves for the experiment (lines 14 and 65), all the calves were breastfed before their age of 30 days.

Point 3:Table 1 Does MR have a vitamin and trace elements content ?

Response 2: Yes, MR contains vitamins and trace elements  including  fat-soluble vitamins A, D and E, water-soluble VB and stress-relieving VC, including VA content of 100 000 ~ 350 000 IU/kg, VE content ≧ 600 IU/kg; trace elements include copper, zinc, iron, etc. We have supplemented them in Table 1(lines 78-82).

Point 4: L96 If rumen fluid is taken only once, how to ensure uniformity?

Response 3: We take rumen fluid only once, but we randomly collected 7 yak calves per group to increase the sample size. In addition, we repeated the procedure three times when collecting rumen fluid, discarding the first time and then mixing the second two times to preserve and measure rumen indicators.

Point5: L143 How can there be multiple comparisons between 2 groups ? How was the mortality or survival anzlyzed ?

Response 4: The statistical methods we mentioned were wrong in the manuscript, so they have deleted it and modified it to "All data were tested for normality using the Sharpiro-Wilk test in R-Studio (3.6.1) and were usually distributed (P > 0.05). Differences of growth performance, and rumen fermentation parameters between the CON and TRT groups were compared using the unpaired t-test" in lines 139-142. We also changed Figure 1 involving survival rates to Table 2 in line 165.

Point 6: TABLE 2  why did rumen fermentation characteristics not include MCP and PH?

Response 5: MCP and pH were not addressed in the experimental design because our main focus was the effect of MR on the early colonization of the rumen by dominant microorganisms. The rumen of calves were still in the developmental stage during the experiment and ruminal microorganisms use lactose as a fermentation substrate at this stage. Which may be related to the direct entry of food from the esophageal groove into the wrinkled stomach.

[1].Wang, H., et al., Effects of compound probiotics on growth performance, rumen fermentation, blood parameters, and health status of neonatal Holstein calves. Journal of Dairy Science, 2022. 105(3): p. 2190-2200. https://doi.org/10.3168/jds.2021-20721

[2].Liu, H., et al., Effects of supplementary concentrate and/or rumen-protected lysine plus methionine on productive performance, milk composition, rumen fermentation, and bacterial population in Grazing, Lactating Yaks. Animal Feed Science and Technology, 2023. 297: p. 115591. https://doi.org/10.1016/j.anifeedsci.2023.115591

[3].Koike, S., et al., Rumen microbiota and its relation to fermentation in lactose-fed calves. Journal of Dairy Science, 2021. 104(10): p. 10744-10752. https://doi.org/10.3168/jds.2021-20225

Point 7: 3.4 relationship or correlation ?

Response 6: It was expressed wrongly, we changed it and  marked in red in the revised paper (Line 248).

Round 2

Reviewer 1 Report

Line 168-169: delete n1 and n2, put them to the sentences of the results.

Line 268: delete In this study, the ADG was 160 g/d.

Line 279: Put “Furthermore,” before calves

Line 292-293: Please reorganize this sentence : In this study, although the differences between the two groups were not significant, estimates were in the concentration range with the highest efficiency (5–25 mg/dL)

Line 301: concentrated ? you meant concentrate?

Line 352-352: Delete “In this study, the effects of two feeding patterns on the growth performance, survival 351 rate, rumen microbiota, and fermentation products of yak calves were analyzed.”

Line 352-358: Please reorganize the sentence. These type of sentences are not summary.

Author Response

Point 1: Line 168-169: delete n1 and n2, put them to the sentences of the results.

Response 1It was deleted as suggested in the revised manuscript (lines 168-169).

Point 2: Line 268: delete In this study, the ADG was 160 g/d.

Response 2: It was deleted as suggested in the revised manuscript (line 276).

Point 3: Line 279: Put “Furthermore,” before calves

Response 3: It was changed as suggested in the revised manuscript (line 287).

Point 4: Line 292-293: Please reorganize this sentence : In this study, although the differences between the two groups were not significant, estimates were in the concentration range with the highest efficiency (5–25 mg/dL)

Response 4: It was changed as suggested in the revised manuscript,and we We added "in NH3-N concentrationsa" and "but the results in two groups" to the sentence,  make the presentation more clearer (lines 301-302).

Point 5: Line 301: concentrated ? you meant concentrate?

Response 5: In the revised manuscript, we found “concentrated" was not very accurate, we have changed it to “ concentrate " (line 311).

Point 6: Line 352-352: Delete “In this study, the effects of two feeding patterns on the growth performance, survival 351 rate, rumen microbiota, and fermentation products of yak calves were analyzed.”

Response 6: It was deleted as suggested in the revised manuscript (line 360).

Point 7: Line 352-358: Please reorganize the sentence. These type of sentences are not summary.

Response 7: In the revised manuscript, we cut out some unnecessary content for clarity, we made changes to the introduction based on your suggestions (Lines 361-366).